# Patch-Wise Semantic Segmentation for Hyperspectral Images via a Cubic Capsule Network with EMAP Features

**Le Sun** [1,2,3] **, Xiangbo Song** [1]**, Huxiang Guo** [1]**, Guangrui Zhao** [1] **and Jinwei Wang** [1,*]

1  School of Computer, Nanjing University of Information Science and Technology (NUIST),
   Nanjing 210044, China; sunlecncom@nuist.edu.cn (L.S.); xiangbosong@nuist.edu.cn (X.S.);
   cs_guohux@nuist.edu.cn (H.G.); cs_zhaogr@nuist.edu.cn (G.Z.)
2  Jiangsu Collaborative Innovation Center of Atmospheric Environment and Equipment Technology, Nanjing
   University of Information Science and Technology (NUIST), Nanjing 210044, China
3  Engineering Research Center of Digital Forensics, Ministry of Education, Nanjing University of Information
   Science and Technology (NUIST), Nanjing 210044, China

**Abstract:** In order to overcome the disadvantages of convolution neural network (CNN) in the current hyperspectral image (HSI) classification/segmentation methods, such as the inability to recognize the rotation of spatial objects, the difficulty to capture the fine spatial features and the problem that principal component analysis (PCA) ignores some important information when it retains few components, in this paper, an HSI segmentation model based on extended multi-morphological attribute profile (EMAP) features and cubic capsule network (EMAP–Cubic-Caps) was proposed. EMAP features can effectively extract various attributes profile features of entities in HSI, and the cubic capsule neural network can effectively capture complex spatial features with more details. Firstly, EMAP algorithm is introduced to extract the morphological attribute profile features of the principal components extracted by PCA, and the EMAP feature map is used as the input of the network. Then, the spectral and spatial low-layer information of the HSI is extracted by a cubic convolution network, and the high-layer information of HSI is extracted by the capsule module, which consists of an initial capsule layer and a digital capsule layer. Through the experimental comparison on three well-known HSI datasets, the superiority of the proposed algorithm in semantic segmentation is validated.

**Keywords:** hyperspectral image classification; morphological attribute profile; cubic convolution; cubic capsule network

## 1. Introduction

In recent years, hyperspectral remote sensing has become an important means of earth observation [1]. Hyperspectral images (HSIs) contain rich spectral and spatial information and have been widely used in agricultural production [2], geological prospecting [3], food safety [4], military target reconnaissance [5] and other important fields. The classification of hyperspectral images plays a very important role in the above fields. With the development of hyperspectral imaging instruments, researchers can obtain HSIs with high spatial resolutions [6,7], which makes HSIs contain more effective information, thus providing great convenience for the development of HSIs segmentation.

At present, there are more and more hyperspectral image classification/segmentation methods based on traditional machine learning techniques [8–12]. Most of them are based on the fusion of spectral and spatial information carried by the images for classification or segmentation [13,14], and representative methods include kernelized support vector machines (k-SVM) [15,16], Markov random fields (MRF) [17], sparse representation (SR) [18,19], morphological transformations (MT) [20,21] and composite kernel or spatial–spectral kernel [22,23]. Although these traditional methods have shown good performance and achieved



appealing results, they cannot fully exploit the deep feature information of hyperspectral images, and it is difficult to significantly improve the classification performance.

Today, deep learning technology has developed rapidly. Because it can independently extract rich deep features, it has outstanding performance in image and video processing, speech recognition and other fields [24–27]. Meanwhile, researchers are also considering the application of deep learning techniques for hyperspectral image processing. Convolutional neural network (CNN) [28], as a classic framework in deep learning, shines in the field of computer vision. In the past few years, classification methods for HSIs based on CNNs have made great progress. Among them, CNN based on two-dimensional convolution has a significant improvement in classification results compared with traditional machine learning based HSI classification methods. For instance, Zhen et al. [29] proposed a multi-scale spatial–spectral CNN, which combines multiple receptive field fusion and multi-scale spatial information for classification. Yu et al. [30] embedded hash semantic features in the CNN framework for hyperspectral image classification. Tun et al. [31] used 2D convolutional layers to learn spatial–spectral features and CNN fully connected layers for classification. Mou et al. [32] proposed 2D–CNN based on spectral attention mechanism, which selectively emphasizes the useful bands and suppresses the useless bands to adaptively calibrate different spectral bands. Gao et al. [33] embedded *t*-distributed random neighborhoods into convolutional neural networks and used 2D–CNN to classify HSIs. Zhang et al. [34] proposed a 2D–CNN that extracts features in different regions of HSIs, showing spectral and spatial context sensitivity. In recent years, the HSI classification/segmentation models based on 2D–CNNs have significantly improved the classification accuracy; however, the two-dimensional convolution mainly lies in the extraction of spatial information while ignoring the spectral information, leaving room for improvement in the accuracies of HSI classification.

CNN based on three-dimensional convolution performs better in hyperspectral image classification and is a good improvement compared on methods based on 2D–CNN. Kanthi et al. [35] proposed a 3D deep feature extraction CNN that simultaneously uses spatial information and spectral information for hyperspectral image classification. Roy et al. [36] proposed a hybrid spectral CNN for HSI classification. This method mainly consists of a spectral 3D–CNN and then a spatial 2D–CNN. Ge et al. [37] proposed an HSI classification method based on 2D–3D CNN and multi-branch feature fusion, that is, 2D–3D CNN extracts image features and a multi-branch neural network performs feature fusion. Zhang et al. [38] proposed an end-to-end 3D lightweight convolutional neural network to solve the problem of hyperspectral image classification with small samples. Sun et al. [39] proposed a segmentation network for hyperspectral image preprocessing to increase the number of pixels of the same class in the block and then used a 3D–CNN-based multi-layer network to classify the hyperspectral image. Zhong et al. [40] proposed a 3D deep residual network based on the combination of spectral residual blocks and spatial residual blocks, that is, firstly 3D convolution was used to extract spectral features in the spectral dimension and then extract spatial features in the spatial domain to classify hyperspectral images. Yu et al. [41] proposed a 2D–3D combined network architecture that performs the classification task of hyperspectral images in which the 2D convolution network is used to extract the spatial features of the hyperspectral image and the 3D convolution network is used to extract the space-spectrum joint features of the hyperspectral image. Those HSIs classification methods based on 3D–CNN have obvious advantages, but when the 3D convolution kernel is large, its extraction of pixel information is too rough, which may easily lead to loss of features and insufficient information representation.

As a new method in deep learning, the capsule network (CapsNet) has outstanding feature extraction capabilities under limited samples due to the application of the idea of locally connected networks [42]. In 2019, Zhu et al. [43] proposed an HSI classification method by combining 3D–CNN and capsule network. In the same year, Paoletti et al. [44] designed a 3D–CNN for joint feature extraction in spatial and spectral domains and combined it with the capsule. The combination of networks improves the classification accuracy

of hyperspectral images and significantly reduces the time complexity of classification. In a word, capsule networks can make up for the shortcomings of ordinary convolutional neural networks that make it difficult to extract finer spatial–spectral features of HSIs patches, because the capsule network uses a vector to express a feature, while CNN uses a numerical value to express it. Furthermore, the method of combining traditional features and deep networks has gradually received attention, such as the CNN network based on 3D Gabor features [45], which can extract richer features than a single CNN network, thereby improving the performance of HSI processing. In theory, as long as the depth and nodes are large enough, the deep neural network can fit any non-linear structure, but this will bring many problems, such as the network not converging or the convergence being too slow. To alleviate this problem, the method of combining traditional features and deep neural network has gradually become a new research line.

In order to overcome the inherent disadvantage of convolutional neural networks and extract richer spatial–spectral features, in this paper, a novel cubic capsule network with extended multi-morphological attribute profile (EMAP) features (termed as EMAP–Cubic-Caps) is used to classify hyperspectral images. Firstly, the principal component analysis (PCA) can only roughly extract the shallow features of the image, especially when fewer principal components are chosen; the information loss is serious, and it brings some other problems. This paper uses EMAP features to extract complex spatial features from hyperspectral images. Then, the EMAP features are used as the input of a cubic convolutional network its main function is to extract the shallow spatial–spectral features. This process is followed by a capsule network, which consists of an initial capsule layer and a digital capsule layer. These two capsule layers are mainly utilized to extract the deep fine features and finally get the classification results. The proposed EMAP–Cubic-Caps is used for hyperspectral image classification tasks to verify its superiorities.

The contributions of our EMAP–Cubic-Caps method are elaborated as follows:

- In theory, the neural network can extract any feature, as long as the network architecture is good enough. However, it is very complicated and time-consuming to design a neural network that can extract a specific geometric structure. Therefore, in this paper, EMAP features are used as the input of the network, which has the advantage of being able to extract rich spatial geometric features well.
- The cubic convolutional network can extract the spatial–spectral features of the hyperspectral image from the three dimensions, which is conducive to making full use of the existing information and improving the classification accuracy.
- The capsule network can further extract more discriminative deep features, such as spectra with the properties of heterogeneity and homogeneity, to better distinguish pixels at the class boundary.

## 2. Deep Capsule Network

Due to the intrinsic structure, CNNs have some shortcomings. For example, the main function of pooling is to retain the key features, reduce the time complexity of the network and make the network invariant to translational transformations. Because of this invariance, it is difficult for CNNs to distinguish the positional relationship of features in the spatial domain, which results in poor ability in distinguishing the fine detailed objects. In 2011, inspired by neuroanatomy and cognitive neuroscience, Hinton proposed the concept of capsules to identify spatial location information. In 2017, Hinton published two papers on the classification of handwritten character sets using the capsule network to achieve the highest classification accuracy.

The capsule network is not composed of scalar neurons but capsules. A capsule is a vector composed of a group of neurons expressed as a vector so that it can represent various features such as the pose and edge of the entity [46]. Multiple capsules to form a hidden layer; the modulus length of the vector in the capsule represents the probability of

classifying the entity, so the squash function is used to specify the modulus length in the interval {0, 1}, as shown in Equation (1).

$$v_j = \frac{\|s_j\|^2}{1 + \|s_j\|^2} \frac{s_j}{\|s_j\|} \tag{1}$$

where $s_j$ represents the capsule input vector, and $v_j$ represents the output vector of the capsule. The squash function does not change the direction of vector $s_j$, only the value of $s_j$, and the greater the value of $\|s_j\|^2$, the closer the value of $\|s_j\|^2/1 + \|s_j\|^2$ is to 1; the smaller the value of $\|s_j\|^2$, the closer the value of $\|s_j\|^2/1 + \|s_j\|^2$ is to 0. This ensures that the learning of features is more stable.

Different from the connection of neurons, the two adjacent layers of capsules are connected in a fully connected way, as shown in Figure 1. The capsule of the $l-1$th layer is fully connected with the capsule of the $l$th layer. $s_i^l$ and $v_i^l$ represent the input vector and output vector of the $i$th capsule of the $l$th layer, respectively. $w_{ij}^l$ and $c_{ij}^l$ respectively represent the weight and coupling coefficient of the connection between the $i$th capsule in the $l-1$th layer and the $j$th capsule in the $l$th layer.

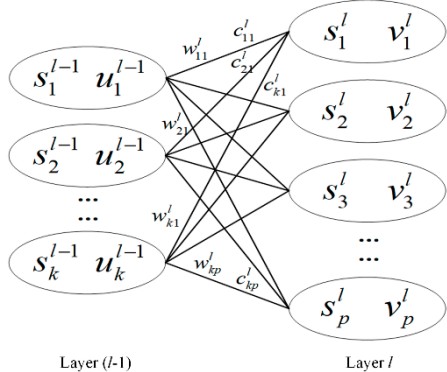

**Figure 1.** Capsule connection mode in the capsule network.

In addition to the input capsule of the first layer, the capsule input vector $s_j^l$ of the subsequent layers needs to be obtained by weighted summation of the prediction vector $\hat{u}_{j|i}^l$. The formula is defined as follows:

$$s_j^l = \sum_i c_{ij}^l \hat{u}_{j|i} \tag{2}$$

$$\hat{u}_{j|i}^l = W_{ij}^l u_i^{l-1} \tag{3}$$

The value of the coupling coefficient $c_{ij}^l$ is updated by the dynamic routing algorithm in the iterative process. The sum of the coupling coefficients of capsule $s_j^l$ and all capsules in the $l$th layer is 1, which is constrained by the softmax activation function [47]. The specific formula is expressed as follows:

$$c_{ij}^l = \frac{\exp(b_{ij}^l)}{\sum\limits_{j=1}^{p} \exp(b_{ij}^l)} \tag{4}$$

where the initial value of parameter $b_{ij}^l$ is 0, which changes during the iteration of the dynamic routing algorithm. It represents the prior probability of the coupling between

the capsule $i$ in the $l-1$th layer and the capsule $j$ in the $l$th layer. The updated formula is as follows:

$$b_{ij}^l \leftarrow b_{ij}^l + \hat{u}_{j|i}^l \cdot v_j^l \tag{5}$$

The dynamic routing algorithm is an excellent iterative algorithm. Its main purpose is to update the coupling coefficient by constantly comparing the degree of consistency between the prediction vector of the previous capsule layer and the output vector of the next capsule layer. It is re-allocated to the prediction vector to coordinate the relationship between the capsule layers so that the output vector of the next capsule layer can find the accurate prediction result. The pseudo-code of the dynamic routing algorithm is elaborated in Algorithm 1 [48].

---

**Algorithm 1.** Pseudo code of dynamic routing algorithm

---

1.  **Initialization**: the number of iterations $k \leftarrow 0$, parameter $b_{ij}^l \leftarrow 0$, total number of iterations $T$.
2.  **While** $k < T$,
3.  Update the coupling coefficient $c_{ij}^l \leftarrow softmax(b_{ij}^l)$.
4.  Update the input vector $s_j^l \leftarrow \sum_i c_{ij}^l \hat{u}_{i|j}^l$.
5.  Update the output vector $v_j^l \leftarrow squash(s_j^l)$.
6.  Update the parameter $b_{ij}^l \leftarrow b_{ij}^l + \hat{u}_{j|i}^l \cdot v_j^l$.
7.  Update the number of iterations $k \leftarrow k + 1$.
8.  **End**
9.  Return the output vector $v_j^l$.

---

Figure 2 shows an architecture of a capsule network, which is composed of a convolutional layer, an initial capsule layer and a digital capsule layer. The convolutional layer uses convolution operations and rectified linear unit (ReLU) activation functions to perform feature extraction on the image, and the output features are used as the inputs of the capsule layer. The initial capsule layer continues to perform convolution operations on the obtained feature maps, converts the local convoluted features into capsules and is fully connected with each capsule in the digital capsule layer. The digital capsule layer has a total of $C$ capsules; here, $C$ represents the number of categories in the data set, and the digital capsules are obtained through a dynamic routing algorithm, where the modulus of the output vector in a capsule represents the probability of being classified into this category.

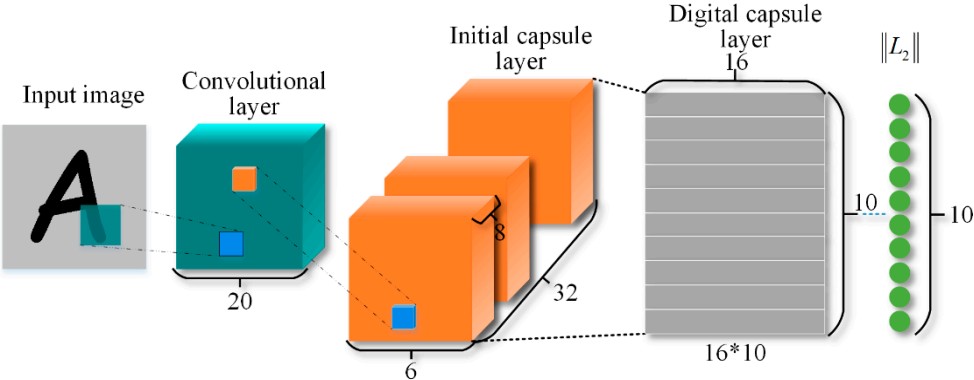

**Figure 2.** Architecture of a capsule network.

## 3. The Proposed Method

### 3.1. Extended Morphological Attribute Profile

Hyperspectral images have hundreds of bands and high spectral resolution. Therefore, it is still a great challenge to analyze and process them. Specifically, due to the Hughes phenomenon, the high dimensionality of the data is a key problem; that is, given a fixed

number of training samples, as the feature dimension increases up to a threshold, the generalization performance of the classifier does not increase but decreases. The threshold mainly depends on the number of training samples used for the classifier. For these reasons, to alleviate the disaster of dimensionality and reduce the amount of calculation, feature extraction is usually used as a preprocessing step. PCA is usually used for hyperspectral image classification tasks. The principle of PCA is to project the data into an orthogonal space so that the eigenvector corresponding to the largest eigenvalue maintains the maximum variance of the data. However, PCA may ignore some important information, especially when few components are retained. Therefore, it is a better choice to use morphological analysis to exploit spatial features of the image after PCA.

Morphological Profile (MP) [49] is a spatial feature extraction operator based on mathematical morphology. The spatial information obtained by using multi-scale analysis can characterize the multi-scale variability of the structure in the image. The disadvantage is that it is difficult to simulate other geometric features in the image. Morphological attribute profile (AP) [50] uses morphological attributes to perform operations under the constraints of specific attribute criteria to obtain a series of attribute refinement profile maps and attribute coarsening profile maps and stack them together. Extended Multiple Morphological Attribute Profile (EMAP) [51,52] uses multiple morphological attributes on the basis of AP algorithm to combine all the obtained profile feature maps for stacking. Compared to the MP algorithm, EMAP is more accurate in representing the spatial information of the image.

For a single band image $f$, its AP is obtained by attribute thickenings and attribute thinnings operation. For a given attribute $A$ and threshold set $B$, the AP algorithm calculates the value of the attribute $A$ of each connected component in the image by comparing with the elements in the set and uses the opening and closing operation to determine whether the attribute thickening ($\varphi$) operation or the attribute thinning ($\gamma$) operation is to be performed. After comparing with all elements in the set, a set of attribute profiles can be obtained.

$$AP(f) = \{\varphi_1(f), \ldots \varphi_n(f), f, \gamma_1(f), \ldots, \gamma_n(f)\} \tag{6}$$

In order to reduce the dimensionality of the hyperspectral image and extract the effective features of the image, PCA is generally used to reduce the dimensionality of the image. Suppose the number of channels after dimensionality reduction is $C$; then, the stacked feature maps after AP operation on the single band image of each channel are called extended morphological attribute profile (EAP).

$$EAPs = \{AP(PC_1), AP(PC_2), \ldots, AP(PC_c)\} \tag{7}$$

where $PC_i$ represents the $i$th principal component. The EMAP feature uses multiple morphological attributes, obtains EAP separately and stacks them together.

$$EMAPs = \{EAP_{a_1}, EAP_{a_2}, \ldots, EAP_{a_n}\} \tag{8}$$

where $a_i$ represents the $i$th morphological attribute. Commonly used morphological attributes include *area*, diagonal length of the external moment of the *area*, moment of *inertia*, *standard deviation*, etc. EMAP has a stronger ability to extract spatial features and has more advantages in extracting the spatial structure of the image.

### 3.2. Cubic Capsule Network with EMAP Features

Aiming at alleviating the difficulty of neural networks to obtain specific spatial structure features in hyperspectral images, inspired by [43], we combined the cubic convolutional network with the capsule network and proposed a cubic capsule network with EMAP features for hyperspectral image classification. At the beginning of the network, the original hyperspectral image was first pre-processed; the high-dimensional hyperspectral data was analyzed by PCA, and the first three principal components were extracted. In addition, three morphological attributes, i.e., the *area*, the diagonal length of the external

moment of the *area*, and the *standard deviation*, were used to extract the EMAP features, and finally, a stacked data cube with 108 feature maps was obtained and was used as the input of the network. We chose an image patch with a size of 15 × 15 × 108 as the size of each training sample, and the batch_size was 100. Figure 3 shows the structure diagram of the proposed EMAP–Cubic-Caps network. The network is divided into three parts, namely cubic convolutional network, initial capsule layer and digital capsule layer.

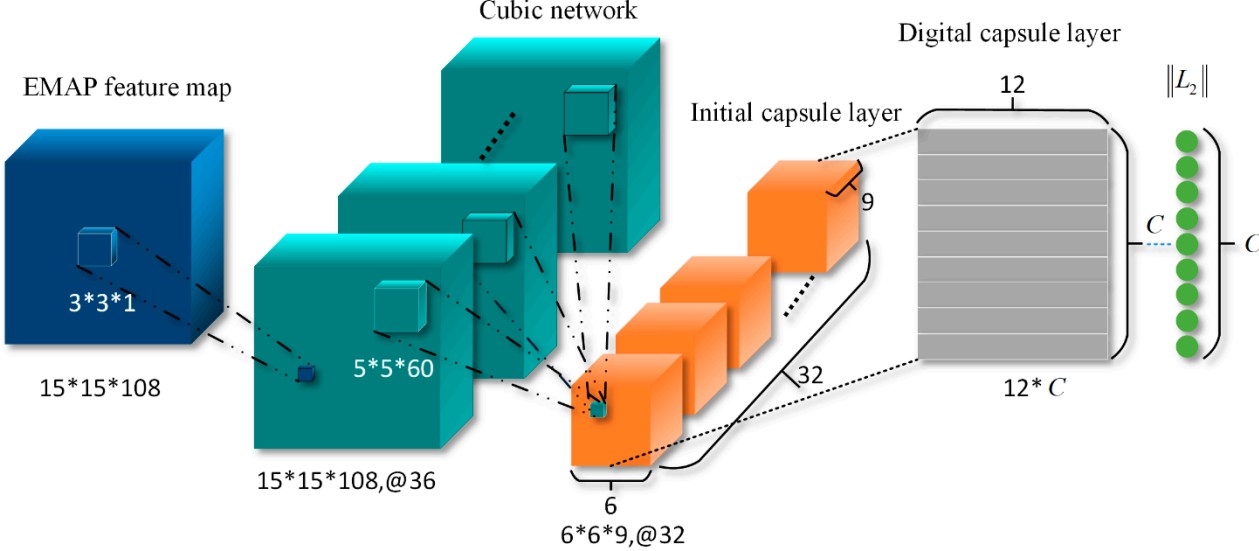

**Figure 3.** Architecture of the proposed EMAP–Cubic-Caps network for HSI classification.

Cubic convolutional network can effectively extract spatial features and spatial–spectral features and is more flexible in training parameters than three-dimensional convolution, and the training speed is faster. The part of cubic convolutional network is shown in Figure 4. That is, in the input 15 × 15 × 108 image patch, the convolution operation is performed on the three planes of the data cube. The size of the convolution kernel of each branch is 3 × 3 × 1, and the number of convolution kernels is 12. The convolution step size is (1, 1, 1), and the convolution is filled with 0. After each convolution layer, the feature map is batch normalized, and the ReLU activation function is performed. Three convolutions are performed on the three branches. After the convolutions, the three branches respectively generate feature maps with sizes (15 × 15 × 108, 12), (15 × 108 × 15, 12) and (108 × 15 × 15, 12). The three data cubes are stacked together to generate a (15 × 15 × 108, 36)-sized feature map, which is used as the input of the initial capsule layer.

The initial capsule layer further combines the features extracted from the cubic convolutional network and encapsulates them into capsules. As shown in Figure 3, the tensor dimension output by the initial capsule layer is (6 × 6 × 9, 32). The input tensor of the capsule layer was conducted by the convolutions operation, where the kernel size is 5 × 5 × 60, the number of kernels is 32, the convolution step size is (2, 2, 8) and those convolutions are not filled by 0. It performs feature integration and obtains a feature map with a size of (6 × 6 × 9, 32). Therefore, we extracted every 9 scalars of the third dimension as vectors and encapsulated them into each capsule.

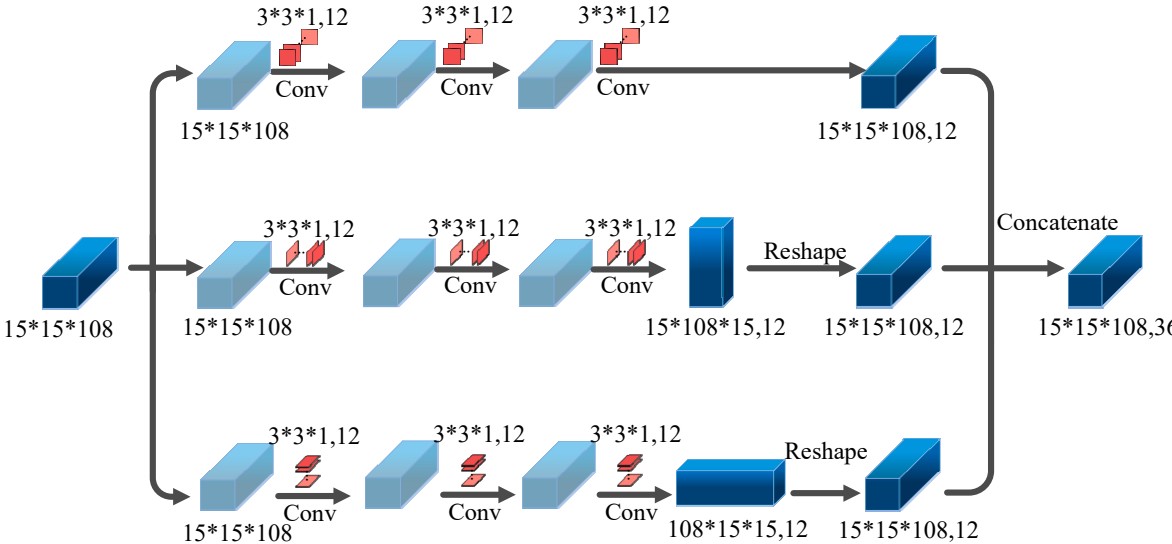

**Figure 4.** Architecture of the cubic convolutional network.

In the digital capsule layer, we set the vector length of each capsule to 12; the number of capsules is *C*, which represents the number of sample categories in the data set. Each capsule represents a type of feature, and each dimension of the vector represents a category of the feature. For example, posture, texture and edge information are each fully connected with all capsules in the initial capsule layer, and the vector in the digital capsule is updated through the dynamic routing algorithm. The number of dynamic routing iterations is 3.

The network proposed in this paper uses Margin loss as the loss function. Since the vector modulus in the capsule of the digital capsule layer represents the probability of being classified into the corresponding class, if the feature is classified as category *k* in the digital capsule layer, then the *k*-th capsule vector modulus will be the largest. The loss function is defined as follows:

$$Loss_k = T_k \max\left(0, m^+ - \|v_k\|\right) + \lambda(1 - T)\max\left(0, \|v_k\| - m^-\right)^2 \tag{9}$$

where $T_k$ is the indicator function, and it is 1 when the samples belong to the category *k*; otherwise, it is 0. $m^+$ represents the upper bound, and $m^-$ represents the lower bound, with values of 0.9 and 0.1, respectively.

The pseudo-code of the proposed EMAP–Cubic-Caps network for hyperspectral image classification is shown in Algorithm 2.

---

**Algorithm 2.** The pseudo code of EMAP–Cubic Caps network

1. **Input**: Hyperspectral data *X* and corresponding label *Y*, the number of iterations $k_1 \leftarrow 0$, $k_2 \leftarrow 0$, total number of iterations $T_1 = 100$, $T_2 = 3$, learning rate $\eta = 0.0003$.
2. Obtain hyperspectral data $X_{EMAP}$ after EMAP feature extraction.
3. Divide $X_{EMAP}$ into training set, verification set and test set, and input the training set and verification set into the cubic convolutional network. The sampling rate is shown in Tables 1–3.
4. **While** $k_1 < T_1$,
5. Perform cubic convolution network.
6. $k_1 \leftarrow k_1 + 1$, algorithm
7. **End**
8. Input the feature map into the initial capsule layer.
9. Connect the initialized capsule to the digital capsule layer and use dynamic routing to update the parameters; see Algorithm 1 for specific steps.
10. Use the trained model to predict the test set.
11. Calculate OA, AA and Kappa.

---

**Table 1.** Labels of Indian Pines data set and the number of training, verification and test samples (each color in the first column stands for one land-cover).

| Category | Class Name | Number of Training Samples | Number of Verification Samples | Number of Test Samples |
|---|---|---|---|---|
| 1 | Alfalfa | 2 | 1 | 43 |
| 2 | Corn-notill | 71 | 36 | 1321 |
| 3 | Corn-mintill | 41 | 21 | 768 |
| 4 | Corn | 11 | 6 | 220 |
| 5 | Grass-pasture | 24 | 12 | 447 |
| 6 | Grass-trees | 36 | 18 | 676 |
| 7 | Grass-pasture-mowed | 1 | 1 | 26 |
| 8 | Hay-windrowed | 23 | 12 | 443 |
| 9 | Oats | 1 | 1 | 18 |
| 10 | Soybean-notill | 48 | 24 | 900 |
| 11 | Soybean-mintill | 122 | 61 | 2272 |
| 12 | Soybean-clean | 29 | 15 | 549 |
| 13 | Wheat | 10 | 5 | 190 |
| 14 | Woods | 63 | 32 | 1170 |
| 15 | Buildings-grass-trees-drive | 19 | 10 | 357 |
| 16 | Store-steel-towers | 4 | 2 | 87 |
| total | | 505 | 257 | 9487 |

**Table 2.** Labels of University of Pavia data set and the number of training, verification and test samples (each color in the first column stands for one land-cover).

| Category | Class Name | Number of Training Samples | Number of Verification Samples | Number of Test Samples |
|---|---|---|---|---|
| 1 | Asphalt | 30 | 15 | 6586 |
| 2 | Meadows | 30 | 15 | 18,604 |
| 3 | Gravel | 30 | 15 | 2054 |
| 4 | Trees | 30 | 15 | 3019 |
| 5 | Painted metal sheets | 30 | 15 | 1300 |
| 6 | Bare Soil | 30 | 15 | 4984 |
| 7 | Bitumen | 30 | 15 | 1285 |
| 8 | Self-Blocking Bricks | 30 | 15 | 3637 |
| 9 | Shadows | 30 | 15 | 902 |
| total | | 270 | 135 | 42,371 |

**Table 3.** Labels of Salinas data set and the number of training, verification and test samples (each color in the first column stands for one land-cover).

| Category | Class Name | Number of Training Samples | Number of Verification Samples | Number of Test Samples |
|---|---|---|---|---|
| 1 | Brocoli_green_weeds_1 | 20 | 10 | 1979 |
| 2 | Brocoli_green_weeds_2 | 20 | 10 | 3696 |
| 3 | Fallow | 20 | 10 | 1946 |
| 4 | Fallow_rough_plow | 20 | 10 | 1364 |
| 5 | Fallow_smooth | 20 | 10 | 2648 |
| 6 | Stubble | 20 | 10 | 3929 |
| 7 | Celery | 20 | 10 | 3549 |
| 8 | Grapes_untrained | 20 | 10 | 11,241 |
| 9 | Soil_vineyard_develop | 20 | 10 | 6173 |
| 10 | Corn_senesced_green_weeds | 20 | 10 | 3248 |
| 11 | Lettuce_romaine_4wk | 20 | 10 | 1038 |
| 12 | Lettuce_romaine_5wk | 20 | 10 | 1897 |
| 13 | Lettuce_romaine_6wk | 20 | 10 | 886 |
| 14 | Lettuce_romaine_7wk | 20 | 10 | 1040 |
| 15 | Vineyard_untrained | 20 | 10 | 7238 |
| 16 | Vineyard_vertical_trellis | 20 | 10 | 1777 |
| total | | 320 | 160 | 53,649 |

## 4. Experiment and Analysis

### *4.1. Experimental Data Set*

In order to validate the effectiveness and generalization of the proposed method, three current well-known hyperspectral data sets, namely the Indian Pines, University of Pavia and Salinas, were employed.

#### 4.1.1. Indian Pines

The Indiana Farm Data Set is a hyperspectral image of a farm in Indiana, USA, taken by the Airborne Visual Infrared Imaging Spectrometer (AVIRIS) in 1992. It has a spatial size of 145 × 145 with a resolution of 20 m per pixel and includes 16 different features and 220 spectral bands with wavelengths ranging from 0.2 micron to 2.4 micron, 20 of which were removed due to the influence of water vapor. Experiments were conducted on the remaining 200 bands. Figure 5a,b are the pseudo-color image and the real label map of the Indian Pines dataset, respectively. Table 1 shows the labels of each class and the number of training, verification and test samples of Indian Pines dataset in the experiment.

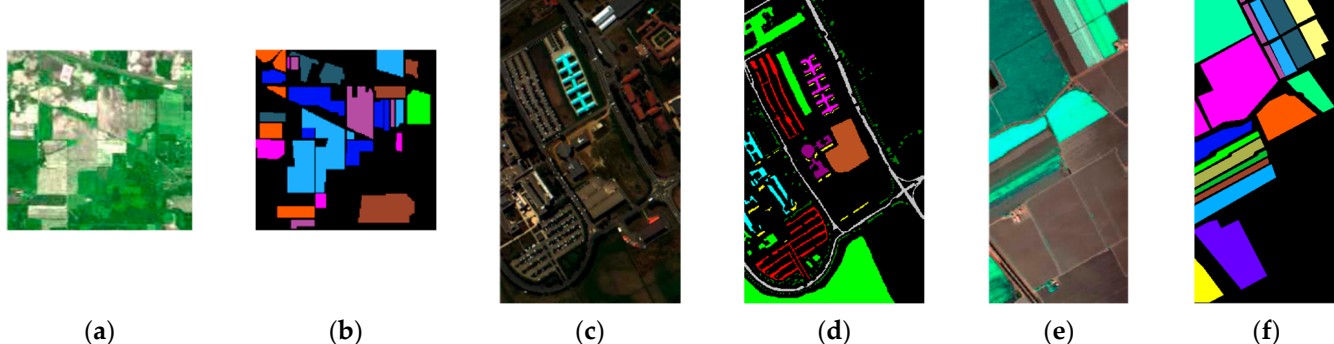

|           (**a**)           |           (**b**)           |           (**c**)           |           (**d**)           |           (**e**)           |           (**f**)           |

**Figure 5.** False color maps and ground-truth maps of three data sets. (**a**,**b**) Indian Pines, (**c**,**d**) University of Pavia, (**e**,**f**) Salinas.

#### 4.1.2. University of Pavia

The data set of the University of Pavia is a hyperspectral image taken by the airborne reflection optical spectral imager (ROSIS-03) at the University of Pavia in Pavia, Italy, in 2003. Its size is 610 × 340, and the spatial resolution is 1.3 m per pixel, including 9 different features, and 115 bands with wavelengths from 0.43 microns to 0.86 microns. Twelve bands were removed due to the influence of noise. Experiments were conducted on the remaining 103 bands. Figure 5c,d are the pseudo-color image and label map of the University of Pavia data set, respectively. Table 2 shows the labels of each class of the University of Pavia data set in the experiment, as well as the training, verification and test samples.

#### 4.1.3. Salinas

The Salinas Valley dataset is a hyperspectral image of the Salinas Valley, California, USA, taken by the Airborne Visual Infrared Imaging Spectrometer (AVIRIS). It has a size of 512 × 217 and a spatial resolution of 3.7 m/pixel. It contains 16 different ground features, including 224 bands. Twenty bands are excluded because of the influence of water vapor. Experiments were conducted on the remaining 204 bands. Figure 5d,e are pseudo-color images and label maps of the Salinas dataset, respectively. Table 3 shows the labels of each class of the Salinas dataset in the experiment and the number of training, verification and test samples.

### *4.2. Experimental Setup*

To verify the superiority of the EMAP–Cubic-Caps network designed in this paper, we tested it on Indian Pines, University of Pavia and Salinas datasets. The number of training, verification and test samples is shown in Tables 1–3. The training samples of the

Indian Pines data set extract a patch with a size of $15 \times 15 \times 108$ centered on the pixel to be classified, and the University of Pavia and Salinas data sets extract a patch with a size of $23 \times 23 \times 108$ centered at the pixel to be classified. The experiments run on Windows 10, and the deep learning platform is Python3.5+ tensorflow1.14.0+ keras2.1.5. The CUP is Intel i7-4790 k with a memory of 24G and the graphics processor of NVIDIA GeForce GTX 1080Ti. Moreover, the overall accuracy (OA), average accuracy (AA) and Kappa coefficient (kappa) are employed as the quantitative indicators to assess the classification performance.

### 4.3. Experiment and Analysis

This section shows the comparison results between the proposed EMAP–Cubic-Caps network and several representative hyperspectral image classification methods to validate the effectiveness. The selected four comparison methods are the support vector machine with EMAP features (EMAP–SVM) [21], the diverse region-based CNN (DR–CNN) [34], the spatial–spectral residual convolutional network (SSRN) [40] and three-dimensional convolutional capsule network based on EMAP preprocessing (3D–Caps) [43]. Among them, for the EMAP–SVM classifier, the hyperspectral data is first preprocessed by PCA, and the first three components are used to extract the EMAP features via three morphological attributes for classification. The DR–CNN method uses a convolutional neural network to extract local features in the upper, lower, left and right directions of the training samples, and merge them with global features for classification. For the SSRN method, a residual network is constructed using 3D convolution kernels with the size of $3 \times 3 \times 128$ to extract the spatial and spectral information of hyperspectral images. For the 3D–Caps method, the EMAP features extracted from the first three principal components are used as the input to the network. 3D–Caps contains two three-dimensional convolutional layers and three capsule layers. The first layer is the initial capsule layer, the last layer is the digital capsule layer and the vectors in the latter two layers of capsules are updated using dynamic routing algorithms. For our proposed method, we named the network without EMAP feature extraction on the original hyperspectral image as Cubic-Caps. It first performs PCA and one-dimensional convolution of the original hyperspectral image to reduce the dimensionality and then uses the cubic convolution network to extract the spectral features and spatial–spectral features, finally sending them to the capsule network for classification.

Table 4 lists the classification accuracies of six algorithms on the Indian Pines dataset. Among them, the result of EMAP–SVM is lower than the accuracy of the neural network-based methods, and the OA is lower than the EMAP–Cubic-Caps proposed in this paper by more than 20%. Compared with the two algorithms DR–CNN and SSRN, the OA of the proposed EMAP–Cubic-Caps also increased up to 98.20%, 2.55% and 8% higher than DR–CNN and SSRN, respectively. Due to the relatively simple part of its convolutional network, the 3D–Caps method has insufficient ability to extract features of hyperspectral images, resulting in its classification accuracy lower than DR–CNN and SSRN methods, but its classification accuracy on the 5th, 12th and 13th categories was higher than other methods, which proves that it has certain advantages. In terms of the average accuracies, the proposed EMAP–Cubic-Caps method obtained the highest AA values, which proves that the classification results of each class obtained by EMAP–Cubic-Caps are satisfactory. Specifically, in the Indian Pines data set, the samples of the 1st, 7th, 9th and 16th categories were extremely unbalanced, which is a big challenge for the classifiers. In the 1st and 7th unbalanced categories with small samples (for both categories, only one sample is selected for training), the proposed EMAP–Cubic-Caps achieved the best class accuracies. In the 9th and 16th categories, the performance of EMAP–Cubic-Caps classifier was relatively general, and its class accuracies of those two categories were only higher than those of the EMAP–SVM method. The reason may be that, in the process of EMAP feature extraction, for very few unbalanced and unevenly distributed samples, the patch-wise-based extraction may weaken the distinguishability of these samples, thus decreasing the class accuracies of those categories. In addition, the Kappa coefficient of the proposed EMAP–Cubic-Caps method was also increased to the highest 0.9765. The algorithm Cubic-Caps without EMAP

feature extraction also obtained good classification results, and its OA exceeded CNN-based method by more than 1%. It is slightly inferior to the proposed EMAP–Cubic-Caps method, which indicates that the capsule network based on cubic convolutional network still has good performance, but the EMAP feature can promote the network to extract richer discriminative features, thereby making the classification accuracy higher. Moreover, among the results of a single class, the proposed EMAP–Cubic-Caps method achieved the highest classification accuracy in the 1st, 4th, 7th, 8th and 13th categories. To summarize, the proposed EMAP–Cubic-CAPs has the best performance among all competitors.

**Table 4.** Classification accuracies of the competing six methods on Indian Pines dataset (the optimal results are shown in bold).

| Category | EMAP–SVM | DR–CNN | SSRN | 3D–Caps | Proposed Methods | |
| --- | --- | --- | --- | --- | --- | --- |
| | | | | | Cubic-Caps | EMAP–Cubic-Caps |
| 1 | 66.04 | 83.33 | 100 | 97.30 | 95.35 | 100 |
| 2 | 75.30 | 96.15 | 96.91 | 95.00 | 95.48 | 98.83 |
| 3 | 40.86 | 96.16 | 95.79 | 93.88 | 96.89 | 98.47 |
| 4 | 51.38 | 98.98 | 98.99 | 99.47 | 98.56 | 99.49 |
| 5 | 90.42 | 95.80 | 95.35 | 96.47 | 96.03 | 96.19 |
| 6 | 97.42 | 95.87 | 96.15 | 99.11 | 95.89 | 97.92 |
| 7 | 66.67 | 92.59 | 96.29 | 73.91 | 65.38 | 100 |
| 8 | 97.11 | 89.98 | 89.78 | 98.25 | 99.34 | 100 |
| 9 | 20.83 | 89.47 | 94.74 | 88.89 | 100 | 75.00 |
| 10 | 71.40 | 92.39 | 92.52 | 89.18 | 93.58 | 97.58 |
| 11 | 98.13 | 98.38 | 98.02 | 74.35 | 98.70 | 98.94 |
| 12 | 80.26 | 92.65 | 92.50 | 98.97 | 95.46 | 94.48 |
| 13 | 88.53 | 100 | 100 | 100 | 100 | 100 |
| 14 | 93.61 | 93.88 | 93.73 | 93.72 | 94.72 | 98.71 |
| 15 | 76.05 | 98.40 | 98.09 | 96.87 | 97.21 | 96.67 |
| 16 | 70.34 | 100 | 100 | 92.55 | 98.85 | 95.35 |
| OA (%) | 76.52 | 95.70 | 95.75 | 90.20 | 96.47 | **98.20** |
| AA (%) | 74.02 | 94.62 | 96.18 | 92.995 | 95.08 | **96.72** |
| Kappa | 0.7383 | 0.9436 | 0.9569 | 0.9015 | 0.9598 | **0.9795** |

Figure 6 shows classification maps of the six algorithms mentioned in this article. It is obvious that the classification map of EMAP–SVM is the most unsatisfactory, and the image has the most noise, because it only uses the combination of EMAP and SVM, and only the shallow features are fused for classification. Two algorithms based on convolutional neural network, i.e., DR–CNN and SSRN, achieved relatively satisfactory results, but there is still some noise in their classification maps. Due to the relatively simple part of the 3D–Caps convolutional network, the classification map obtained by it is poorer than DR–CNN and SSRN methods. The proposed Cubic-Caps and EMAP–Cubic-Caps have the least noise on the whole maps and get the highest degree of consistency with the distribution of land-covers, especially for the proposed EMAP–Cubic-Caps method.

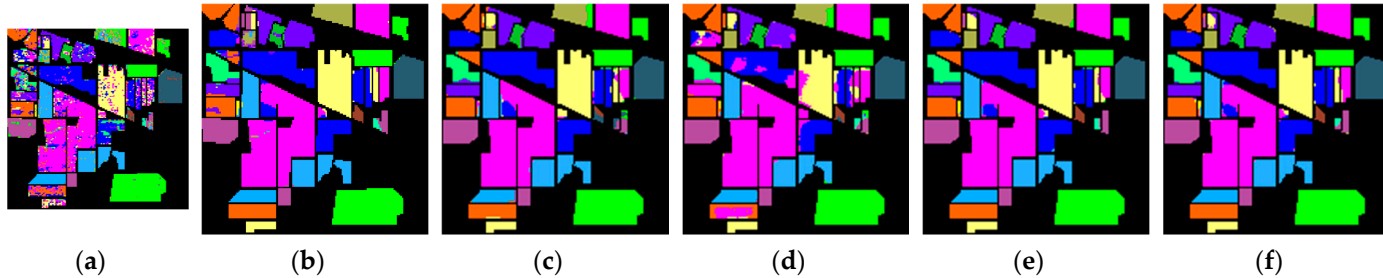

| (a) | (b) | (c) | (d) | (e) | (f) |

**Figure 6.** Classification maps of six algorithms on Indian pines dataset. (**a**) EMAP–SVM, (**b**) DR–CNN, (**c**) SSRN, (**d**) 3D–Caps, (**e**) Cubic-Caps, (**f**) EMAP–Cubic-Caps.

Benefitting from the spatial geometric features of EMAP, the results of EMAP–Cubic-Caps on the Alfalfa and Buildings-grass-trees-drive classes are significantly better than other methods, which proves that the features of the vector encapsulated into capsules better represent the information of the land cover, so the classification results are better than other competitors. It can be drawn from the above that the conclusions obtained from the classification result maps are highly consistent with the results of the quantitative evaluation.

Table 5 shows the classification accuracies of six algorithms on the University of Pavia dataset. Among them, the results of EMAP–SVM are not satisfactory, and the OA is lower than the methods based on the convolutional neural network by about 20% and lower than the proposed EMAP–Cubic-Caps method by 22.36%. Compared with the two convolutional neural network methods, i.e., DR–CNN and SSRN methods, the OAs of our EMAP–Cubic-Caps method increased from 94.40% and 95.15% to 98.81%, an increase of more than 3%. The 3D–Caps algorithm based on the three-dimensional convolutional capsule network had only two convolutional layers because of its convolutional network part. The ability to extract low-level features of HSI is poor, so the OA of it was only 88.30%. In terms of the average accuracy (AA), the proposed EMAP–Cubic-Caps method achieved the highest value of 98.49%, which validates that the algorithm in this paper has great discrimination in classifying most of the instances. In addition, for class-accuracies, the EMAP–Cubic-Caps method basically achieved the best performance. In particular, in the University of Pavia dataset, the 1st, 2nd, 3rd, 5th, 6th, and 8th category, the proposed EMAP–Cubic-Caps method achieved the highest accuracy. Moreover, the Kappa coefficient of EMAP–Cubic-Caps method increased from the other highest 0.9364 (SSRN method) to 0.9842. It indicates that the proposed classifier has better performance in inner-class consistency. Even though the proposed Cubic-Caps method does not perform EMAP feature extraction, it also obtained good results, which are slightly inferior to the results of EMAP–Cubic-Caps classifier. Again, it further illustrates the advantages of EMAP–Cubic-Caps method in processing hyperspectral image classification tasks.

**Table 5.** Classification accuracies of the competing six methods on University of Pavia dataset (the optimal results are shown in bold).

| Category | EMAP–SVM | DR–CNN | SSRN | 3D–Caps | Proposed Methods | |
| --- | --- | --- | --- | --- | --- | --- |
| | | | | | Cubic-Caps | EMAP–Cubic-Caps |
| 1 | 71.29 | 98.63 | 98.70 | 98.41 | 98.86 | 99.63 |
| 2 | 75.75 | 98.34 | 98.46 | 97.49 | 99.75 | 99.76 |
| 3 | 72.97 | 99.30 | 98.96 | 79.20 | 99.10 | 99.85 |
| 4 | 91.80 | 90.73 | 91.68 | 99.86 | 98.24 | 98.98 |
| 5 | 99.33 | 99.32 | 99.39 | 99.77 | 99.92 | 100 |
| 6 | 71.55 | 74.53 | 78.33 | 58.82 | 93.57 | 95.15 |
| 7 | 87.60 | 99.61 | 99.45 | 90.60 | 93.92 | 95.73 |
| 8 | 67.29 | 97.15 | 97.55 | 87.11 | 95.57 | 97.59 |
| 9 | 99.31 | 100 | 100 | 100 | 99.45 | 99.77 |
| OA (%) | 76.45 | 94.30 | 95.15 | 88.30 | 98.15 | **98.81** |
| AA (%) | 81.88 | 95.29 | 95.84 | 90.14 | 97.60 | **98.49** |
| Kappa | 0.6985 | 0.9254 | 0.9364 | 0.8493 | 0.9755 | **0.9842** |

Figure 7 illustrates the classification maps of six algorithms on the University of Pavia dataset. Among them, the classification results of the EMAP–SVM algorithm still contain a lot of noise. The other five algorithms based on neural networks have results in classifying the corresponding land-covers. Among those five network-based classifiers, the performance of 3D–Caps is slightly worse. Many pixels belonging to the Meadows category are classified into the Bare Soil category. The classification map of the proposed EMAP–Cubic-Caps method has the least noise and better restores the distribution of corresponding land-covers. In addition, the results of EMAP–Cubic-Caps on the Self-Blocking Bricks and

Bare Soil classes are significantly better than other methods. It proves that the ability of vector-encapsulated capsules to represent land cover information is stronger than that of scalar neurons, which further illustrates the advantages of the capsule network.

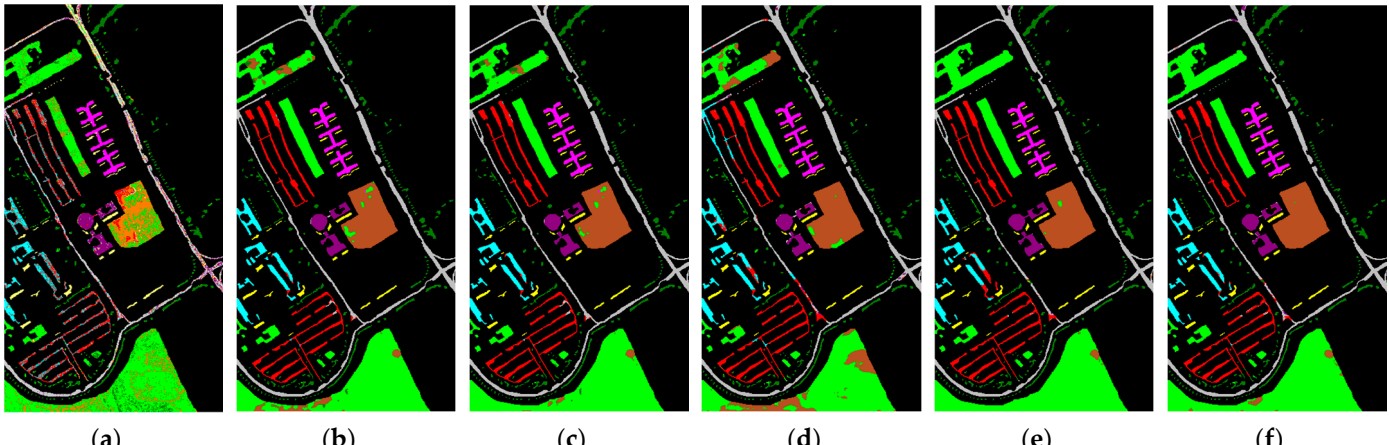

**Figure 7.** Classification maps of six algorithms on University of Pavia dataset. (**a**) EMAP–SVM, (**b**) DR–CNN, (**c**) SSRN, (**d**) 3D–Caps, (**e**) Cubic-Caps, (**f**) EMAP–Cubic-Caps.

Table 6 shows the quantitative classification accuracies of six algorithms on the Salinas dataset. Similarly, the results of EMAP–SVM are far from those neural network-based algorithms, especially in the categories of Grapes_untrained and Vineyard_untrained. The performance of EMAP–SVM was poor, and the overall accuracy (OA) was the lowest. Compared with the two algorithms based on CNN, i.e., DR–CNN and SSRN methods, the proposed EMAP–Cubic-Caps classifier improved OA indicators by 5.4% and 3.11%, respectively, and increased by 2.6% and 2.1% on AA indicators, respectively. Based on the 3D–Caps method of the 3D convolutional capsule network, the OA of EMAP–Cubic-Caps increased by nearly 10%, and the AA increased by nearly 5%. Moreover, the Kappa coefficient was also improved. It verifies that the proposed EMAP–Cubic-Caps method can achieve stable classification results with more categories and has better generalization performance in the classification of multi-category land cover. The experimental results of the proposed Cubic-Caps method are slightly worse than the proposed EMAP–Cubic-Caps classifier, which again validates that, with the help of EMAP features, the accuracy of classification can be effectively improved. Through comparison, it is obvious that the proposed Cubic-Caps and EMAP–Cubic-Caps method have superior advantages.

Figure 8 shows the classification maps of the above-competing methods on the Salinas data set. Among them, the classification result map based on the EMAP–SVM algorithm has the worst performance. There is also some noise in the DR–CNN and SSRN methods. In the classification results of the Fallow and Grapes_untrained classes, there is more noise, and the classification maps of other classes perform better. 3D–Caps performs a little bit worse than DR–CNN and SSRN methods, and the noise is more than other neural-network-based algorithms. The classification map of the proposed EMAP–Cubic-Caps method has the least noise and better restores the real distribution of land-covers. The results on the two classes of Fallow and Grapes_untrained are significantly better than other methods. It again validates that the ability of using vector-encapsulated capsules to represent land cover information is stronger than that of scalar neurons. It also once again proves the advantages of the proposed algorithm.

**Table 6.** Classification accuracies of the competing six methods on Salinas dataset (the optimal results are shown in bold).

| Category | EMAP–SVM | DR–CNN | SSRN | 3D–Caps | Proposed Methods | |
| --- | --- | --- | --- | --- | --- | --- |
| | | | | | Cubic-Caps | EMAP–Cubic-Caps |
| 1 | 92.90 | 100 | 100 | 100 | 100 | 100 |
| 2 | 92.77 | 96.89 | 100 | 100 | 100 | 100 |
| 3 | 93.49 | 98.78 | 97.55 | 94.98 | 100 | 100 |
| 4 | 87.87 | 95.84 | 94.86 | 95.34 | 98.60 | 99.34 |
| 5 | 90.61 | 99.92 | 99.49 | 98.43 | 99.96 | 99.96 |
| 6 | 85.51 | 99.85 | 99.90 | 100 | 100 | 100 |
| 7 | 89.12 | 100 | 100 | 99.58 | 100 | 99.97 |
| 8 | 51.43 | 93.24 | 93.73 | 83.76 | 99.66 | 98.90 |
| 9 | 93.61 | 98.68 | 99.50 | 99.36 | 99.77 | 99.91 |
| 10 | 66.20 | 88.68 | 95.38 | 91.76 | 97.00 | 98.44 |
| 11 | 87.70 | 100 | 93.97 | 94.26 | 99.07 | 99.89 |
| 12 | 82.91 | 99.95 | 100 | 99.89 | 100 | 100 |
| 13 | 77.19 | 98.22 | 100 | 99.53 | 100 | 100 |
| 14 | 71.08 | 99.71 | 94.32 | 97.79 | 99.69 | 98.68 |
| 15 | 64.01 | 73.81 | 83.06 | 60.50 | 86.57 | 99.34 |
| 16 | 94.78 | 99.94 | 100 | 94.47 | 99.88 | 96.88 |
| OA (%) | 76.74 | 93.15 | 95.44 | 88.95 | 97.39 | **98.55** |
| AA (%) | 82.57 | 96.47 | 96.98 | 94.35 | 98.70 | **99.08** |
| Kappa | 0.7539 | 0.9239 | 0.9493 | 0.8774 | 0.9709 | **0.9838** |

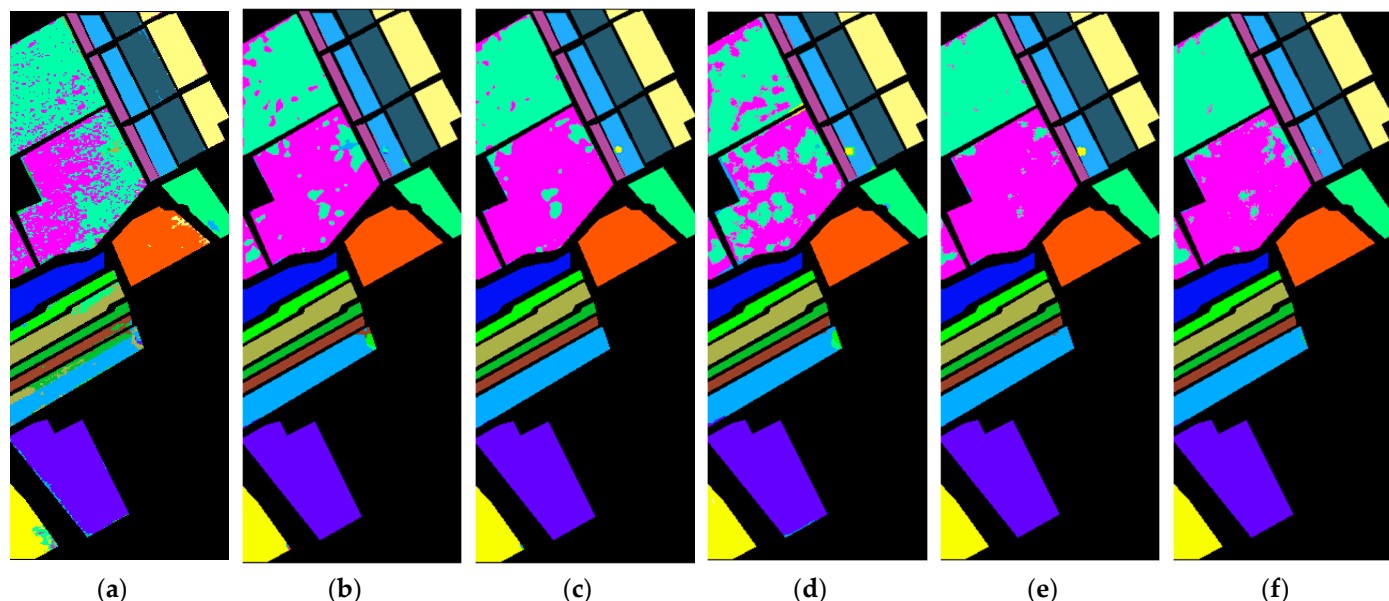

**Figure 8.** Classification maps of six algorithms on Salinas dataset. (**a**) EMAP–SVM, (**b**) DR–CNN, (**c**) SSRN, (**d**) 3D–Caps, (**e**) Cubic-Caps, (**f**) EMAP–Cubic-Caps.

## 5. Discussion

Table 7 shows the comparison of the time complexity of training and testing of different algorithms on the three data sets. Among them, the DR–CNN based on two-dimensional convolutional network has a more complex network structure and consumes the most training and testing time. The SSRN of the residual network has a good performance in time complexity. The 3D–Caps based on the 3D convolutional capsule network is relatively simple because of its convolutional network part, so it consumes the shortest training and testing time. Although the time complexity of Cubic-Caps is higher than that of 3D–Caps, it has a better performance on classification accuracy than the latter.

**Table 7.** Training and testing time complexity of each algorithm on different data sets.

| Competing Methods | | IN | UP | Salinas |
|---|---|---|---|---|
| DR–CNN [34] | Train. (min) | 7.86 | 7.62 | 7.74 |
| | Test.(s) | 55.71 | 97.19 | 281.25 |
| SSRN [40] | Train. (min) | 2.62 | 1.41 | 2.06 |
| | Test.(s) | 4.49 | 12.24 | 25.91 |
| 3D–Caps [43] | Train. (min) | 1.52 | 1.15 | 1.38 |
| | Test.(s) | 4.44 | 11.46 | 17.56 |
| Cubic-Caps | Train. (min) | 1.94 | 1.22 | 1.84 |
| | Test. (s) | 5.43 | 14.36 | 27.64 |
| EMAP–Cubic-Caps | Train. (min) | 1.76 | 1.29 | 1.50 |
| | Test.(s) | 4.21 | 13.33 | 25.47 |

By the ablation experiment (that is, experiments of Cubic-Caps and EMAP–Cubic-Caps methods), Cubic-Caps has high time complexity due to the use of raw data for training. In contrast, the proposed EMAP–Cubic-Caps method can obtain satisfactory accuracy under good time complexity, which fully proves the superiority of the EMAP–Cubic- Caps method.

Figure 9a shows the comparison of the classification overall accuracy over the number of convolutional kernels in each layer of the Cubic Network part of the EMAP–Cubic-Caps method. It is clear that, when the number of convolution kernels is 3, the result is not satisfactory, and the network′s ability to extract image information is insufficient. When the number of convolution kernels is selected as 6 and 12, the classification accuracy continuously improves. When the number of convolution kernels is selected as 24, the classification accuracy tends to converge. Therefore, we used 12 in each convolution layer to learn the characteristics of the image and it results in optimal results.

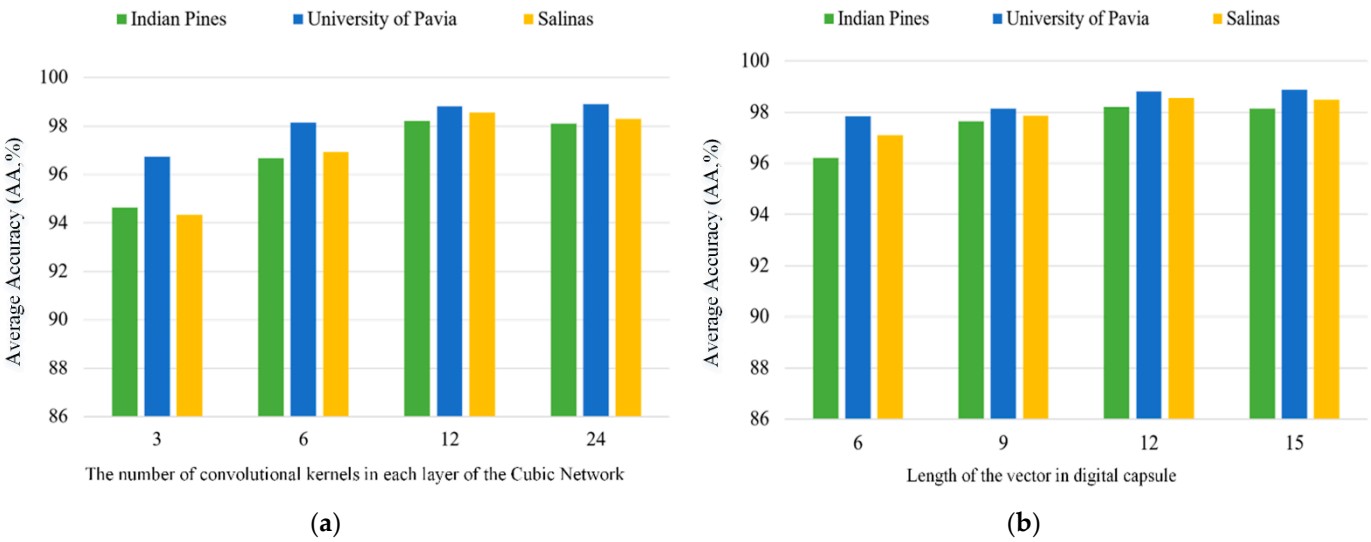

(**a**)   (**b**)

**Figure 9.** (**a**) Comparison of classification average accuracy over the number of convolutional kernels in each layer of the cubic convolutional network. (**b**) Comparison of classification average accuracy over the vector modulus of the digital capsule.

Figure 9b shows the comparison of the OA over the length of vector of each digital capsule in the digital capsule layer. It can be observed that, when the length of the vector is set to 6 and 9, the classification OA continuously rises. When the length is set to 12 and 15, the classification accuracy changes from rising to converging. In order to ensure the

efficiency of the algorithm proposed in this chapter, we chose 12 as the length of the vector in the digital capsule layer.

## 6. Conclusions

In the paper, a cubic capsule network with EMAP features (EMAP–Cubic-Caps) is proposed to classify hyperspectral images, which can effectively alleviate the defect of insufficient spatial–spectral feature extraction of hyperspectral images by most convolutional neural networks. The EMAP–Cubic-Caps network is composed of EMAP feature extraction, a cubic convolutional network, an initial capsule layer and a digital capsule layer. EMAP first extracts three geometric structural features from the three principal components of the original hyperspectral image. The function of the cubic convolutional network is to extract the spatial–spectral features of the image from three planes of the cube. The two capsule layers further use vector-encapsulated capsules to extract richer and more accurate deep features, thereby improving the classification accuracy of HSIs. Through experimental comparison, it is verified that the performance of the proposed EMAP–Cubic-Caps method is better than several state-of-the-art CNN-based methods. In addition, compared with 3D–Caps, the proposed EMAP–Cubic-Caps method has improved significantly in all terms of accuracies. Specifically, the advantage of the proposed EMAP–Cubic-Caps method is that it can fully extract geometric morphological features and integrate them into the capsule network to better express the features of the ground cover. It has a good performance in classification for the scenes with fewer samples and rich geometric details (for example, a local area with rich detailed information and diverse shapes). In the ablation experiment, that is, in comparison with the 3D–Caps method, the proposed EMAP–Cubic-Caps network fully extracted the low-level features of the hyperspectral image before the capsule layer, which verifies that the performance of the model trained using EMAP features is better than the original data.

**Author Contributions:** Conceptualization, X.S. and L.S.; Funding acquisition, L.S. and J.W.; Investigation, H.G. and G.Z.; Methodology, L.S. and X.S.; Software, X.S.; Supervision, L.S. and J.W.; Validation, X.S. and L.S.; Visualization, L.S., H.G. and X.S.; Writing—original draft, L.S., X.S., H.G., G.Z. and J.W.; All authors have read and agreed to the published version of the manuscript.

**Funding:** This study was supported by the National Natural Science Foundation of China (61971233, 62076137, U1831127), the Henan Key Laboratory of Food Safety Data Intelligence (KF2020ZD01), and the Postgraduate Research & Practice Innovation Program of Jiangsu Province (KYCX21_1004).

**Institutional Review Board Statement:** Not applicable.

**Informed Consent Statement:** Not applicable.

**Data Availability Statement:** Not applicable.

**Conflicts of Interest:** The authors declare no conflict of interest.

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
