# Peer review of "Patch-Wise Semantic Segmentation for Hyperspectral Images via a Cubic Capsule Network with EMAP Features"

_remotesensing, doi:10.3390/rs13173497_

Round 1
Reviewer 1 Report
This paper presents interesting results for a patch-wise semantic segmentation. Besides results accuracy, the proposed approaches seams to have a good ability in taking decisions along the borders between two different regions. That quality is very relevant, and since it is quite uncommon in patch-wise approaches, it should be more exploited through out the paper.
Despite its qualities, I have some questions and notes about your work.
First, I would say that the paper title should bring the words “patch-wise semantic segmentation”, in accordance to the taxonomy proposed by Volpi and Tuia (2017) in IEEE Geoscience and Remote sensing letters 55 (2) https://doi.org/10.1109/TGRS.2016.2616585 .
On page 3, lines 118 to 129, authors present EMAP-Cubic-Caps highlights. Please, take a look at the first item, and consider presenting it more concisely.
I would say it may be positive changing the formalism used on algorithms 1 and 2 to a more similar to the ones used in computer science. Look at this link (https://farm5.staticflickr.com/4747/25697721187_0986b0d994_b.jpg) as a reference.
On page 11, line 346 it is seems be “CPU” instead of “CUP”.
In the section 4.3, I would say that it is necessary justifying that choice for such methods (EMAP-SVM, DR-CNN, SSRN and 3D-Caps) used in the comparison. I would say that it is also desirable describing the main advantages and disadvantages of such methods. It is also necessary presenting more details about the training procedures as well the exact configurations of such methods.
In figures 5, 6 and 7, I would say that it is desirable bring the ground truth images to these figures.
The paper is lacking some more deep analysis about the performance class-to-class (considering the specificities of each database) obtained. As far as I am concerned, results analysis must be improved. Please, bring a kind of ablation subsection looking forward facing performance class to class and methods’ performance expectations.
Results presented in Figure 9, should be the first ones to be presented in the paper, since it can be used for fine tuning the proposal hyperparameters. However, I disagree with the use of overall accuracy in this evaluation, since it may mask a poor performance in minoritarian classes. I suggest you use AA, for instance.
Author Response
see the attached file for details

Reviewer 2 Report
18/08/2021
Dear authors,
In the manuscript Hyperspectral Classification Via a Cubic Capsule Network with EMAP Features an hyperspectral image (HIS) classification model based on EMAP (Extended Multiple Morphological At-tribute Profiles) features and cubic capsule network (EMAP-Cubic-Caps) was proposed. EMAP features can effectively extract various attributes profile features of entities in HSI, and the cubic capsule neural network can effectively capture complex spatial features with more details. Through the experimental comparison on three well-known HSI datasets, the superiority of the proposed algorithm in classification is validated.
General comments
The theme is wary actual and interesting. My personal assessment on the work is positive, owing to the relevant, timely and clarity of the study, as well as the focus of analysis. The study is interesting and experimental results may have great usability in the hyperspectral remote sensing.
However, I think the Introduction should be refined with a better overview of the situation in the area, especially as far as your method is concerned.
I also suggest that you revise the Conclusion according to the comment below. In the Conclusion, you do not need to explain how you did something (you did it in the manuscript), but interpret the results and emphasize what your proposed method is better at.
Specific comments (are in the manuscript)
- Line 110 - This paper uses: nothing! However, the experts who conducted the research (which you write about in this manuscript) used ‘the extended multi-morphological attribute profile (EMAP)’. Please be precise when writing texts like this.
- Line 523-539 - The statements in the Conclusion should be supported by the results you obtained in the research, and not just descriptively. Conclusion should be a detailed final interpretation of all the results obtained in the research, and highlight the novelty you introduce.
Best regards

Author Response
see the attached file for details

Reviewer 3 Report
In this paper the performances of a novel cubic capsule network with EMAP features (EMAP-Cubic-Caps) was evaluated for classifying hyperspectral images. And then the advantages were confirmed from the two viewpoints extracting rich spatial geometric features and processing time.
I list here below some comments to improve the presentation, but in general I would say that the manuscript requires minor revisions only.
LL.94-96
You should cite Sabour et al (2017).
Sabour, S., Frosst, N., and Hinton, G.E. Dynamic Routing Between Capsules, Advances in Neural Information Processing Systems 30 (NIPS2017).
You said that the evaluations were based on OA, CA, AA and Kappa.
I would suggest the authors give more exact descriptions of the assessment method used in the paper.
Figure 5-8
Could you add scale bars?
Why did EMAP-Cubic-Caps possess the relatively low accuracy for identifying Oats?
The sample sizes of Alfalfa, Grass-pasture-mowed and Oats were quite small. Is there sample data available to verify the accuracy and stability of the models?
Author Response
see the attached file for details
